Exosomal non-coding RNAs: key molecules in the diagnosis and treatment of coronary artery disease

Ma Cuixue 1 2
Gao Xinyu 1 2
Li Kongwei 1 2
Liu Yin 1 2
He Yuan 1 2
Zhang Yang 3
Lei Wei leiwei2006@126.com 1 2 4
1 Guangdong Provincial Engineering Technology Research Center for Molecular Diagnosis and Innovative Drugs Translation of Cardiopulmonary Vascular Diseases, University Joint Laboratory of Guangdong Province and Macao Region on Molecular Targets and Intervene, Affiliated Hospital of Guangdong Medical University , Zhanjiang , China
2 Laboratory of Cardiovascular Diseases, Affiliated Hospital of Guangdong Medical University , Zhanjang , China
3 Affiliated Hospital of Guangdong Medical University , Zhanjiang , Guangdong , China
4 Precision Medicine Center, Affiliated Hospital of Guangdong Medical University , Zhanjianig , China
Zhang Xin
Electronic publication date: 2025 Jun 10
Publication date: 2025
Volume: 13
Electronic Location ID: e19352
Received 2024 Mar 7; Accepted 2025 Mar 31
Copyright: ©2025 Ma et al.
Copyright year: 2025
Copyright holder: Ma et al.
License: This is an open access article distributed under the terms of the Creative Commons Attribution License, which permits unrestricted use, distribution, reproduction and adaptation in any medium and for any purpose provided that it is properly attributed. For attribution, the original author(s), title, publication source (PeerJ) and either DOI or URL of the article must be cited.
License URL: https://creativecommons.org/licenses/by/4.0/

Keywords: Coronary artery disease, Exosomes, Non-coding RNA

Funding: National Natural Science Foundation of China 81700269 Discipline Construction Project of Guangdong Medical University 4SG21233G Natural Science Foundation of Guangdong Province 2019A1515011925 Key platform of Department of Education of Guangdong Province 2021LSYS007 Zhanjiang Science and Technology Development Special Funding Competitive Allocation Project 022E05011, 2022A01196, 2021A05158 This work was supported by National Natural Science Foundation of China (81700269), Discipline Construction Project of Guangdong Medical University (4SG21233G), Natural Science Foundation of Guangdong Province (2019A1515011925), Key platform of Department of Education of Guangdong Province (2021LSYS007), and Zhanjiang Science and Technology Development Special Funding Competitive Allocation Project (2022E05011, 2022A01196, 2021A05158). The funders had no role in study design, data collection and analysis, decision to publish, or preparation of the manuscript.

==============================
Coronary artery disease (CAD) is a leading cause of mortality worldwide. As aging populations grow and lifestyle changes, the incidence of CAD is escalating. Traditional biomarkers for CAD diagnosis, such as creatine kinase-muscle brain (CK-MB), troponins, and n-terminal pro b-type natriuretic peptide (NT-proBNP), are influenced by age, sex, the presence of chest pain, and renal function levels. However, since these biomarkers are also detected in many other diseases such as heart failure, chronic renal failure, pulmonary embolism, or septic shock, explore and identification of novel unique biomarkers for CAD are of clinical significance. Exosomes containing non-coding RNAs, proteins, and lipids can serve as biomarkers and regulators for regulate various biological processes. Exosomal non-coding RNAs have been identified as risk factors for CAD and pivotal elements in cellular functions, making them significant candidates for CAD diagnosis and prognosis. This review elaborates on the current understandings of CAD, highlights the important roles of exosomal non-coding RNAs in CAD diagnosis and treatment, and concludes with future perspectives.

Introduction

Coronary artery disease (CAD) accounts for over 8 million deaths annually and is one of the leading causes of mortality worldwide (Çakmak & Demir, 2020), imposing significant burdens on healthcare systems and family economies (Li et al., 2023). In the United States, while the number of certain CAD risk factors, such as smoking, hypertension, dyslipidemia, and lack of exercise, is decreasing, others like aging, obesity, diabetes, and insulin resistance are increasing (Duggan et al., 2022; Lu et al., 2023). Based on clinical phenotypes, CAD can be classified into chronic coronary syndromes (stable angina, ischemic cardiomyopathy, and silent ischemia), acute coronary syndromes (unstable angina (UA), non-ST-elevation myocardial infarction (NSTEMI), and ST-elevation myocardial infarction (STEMI)) (Malakar et al., 2019).

The coronary artery wall is composed of three including the intima, media, and adventitia (Krüger-Genge et al., 2019; Sandoo et al., 2010). The endothelium of blood vessels releases vasoactive substances that regulate vascular tone and remodeling of the vascular wall (Bonetti, Lerman & Lerman, 2003). When coronary endothelial cells are damaged or encounter risk factors such as dyslipidemia, hypertension, hyperglycemia-related oxidative products, or pro-inflammatory cytokines produced by excess adipose tissues, the endothelium can activate and trigger the expression of leukocyte adhesion molecules (Libby & Theroux, 2005). These adhesion molecules include vascular cell adhesion molecule-1, intercellular adhesion molecule-1, E-selectin, and P-selectin (Matsuzawa & Lerman, 2014). Low-density lipoprotein (LDL) enters the intima through damaged endothelium and is oxidatively modified to oxidized LDL cholesterol (oxLDL-C) (Zhang, Sessa & Fernández-Hernando, 2018). Endothelial and smooth muscle cells secrete monocyte chemoattractant protein-1 and macrophage colony-stimulating factor, which promote monocyte chemotaxis, adhesion, and differentiation into macrophages. These macrophages engulf oxLDL-C via scavenger receptors and transform into foam cells, forming the earliest lesion of lipid streaks (Liao, 1998). The evolution from a lipid streak to fibro-fatty lesions and fibrous plaques is considered a cytokine-mediated inflammatory response (Arbustini & Roberts, 1996; Rognoni et al., 2015). Anatomically, plaques that do not rupture, erode, ulcerate, or form thrombi are stable, whereas plaques rupture can lead to acute cardiovascular events (Ahmed, Bittl & Braunwald, 1993; Rosenschein et al., 1994) (Fig. 1).

Figure 1 Risk factors and pathogenesis of CAD.

Risk factors CAD include age, smoking, diabetes, hypertension, insulin resistance, dyslipidemia, and lack of exercise. When coronary endothelial cells are damaged, oxidation products and pro-inflammatory cytokines released by certain risk factors enter the endothelium. Vascular endothelial cells and smooth muscle cells secrete monocyte chemotactic protein-1 and macrophage colony-stimulating factor in response to oxidative products and proinflammatory cytokines. These factors promote monocyte chemotaxis, adhesion, and differentiation into macrophages, phagocytosis of oxLDL-C via scavenger receptors, and transformation into foam cells to form the earliest lesion lipid streaks. LDL enters the intima through damaged endothelium and is oxidatively modified to oxLDL-C. Lipid streaks gradually evolve into fibrous plaques, and when the plaque ruptures it leads to an acute cardiovascular event. Abbreviations: LDL. Low-density lipoprotein; oxLDL-C, oxidized LDL cholesterol. Copyright: Figdraw.

In clinical settings, the primary imaging tests for diagnosing CAD are coronary computed tomography angiography and coronary angiography (Leipsic & Tzimas, 2023). To date, the CAD assessment primarily relies on indicators such as the presence or absence of angina during rest or exercise, electrocardiogram changes, CK-MB, troponins, and NT-proBNP (Zhang et al., 2020). These biomarkers aid in individualized treatment of patients, yet there remains debate over which specific biomarker should be the standard (Pei et al., 2023; Tokgozoglu, Morrow & Nicholls, 2023). Some patients with UA may exhibit normal electrocardiograms and normal troponin levels without myocardial ischemia, where preliminary diagnosis depends on the patient’s clinical history and the clinician’s judgment (Amsterdam et al., 2014). Pharmacotherapy for CAD includes beta-blockers, calcium channel blockers, ACEI/ARB, antiplatelet agents, and statins. For patients with complex lesions, treatment primarily involves coronary artery bypass grafting, percutaneous coronary intervention and lipid-lowering therapy (Duggan et al., 2022; Jia, Liu & Yuan, 2020). However, traditional biomarkers, influenced by age, sex, presence or absence of chest pain, and renal function levels, are also detected in many other diseases like heart failure, chronic renal failure, pulmonary embolism, and septic shock. Thus, the exploration and identification of new unique biomarkers for CDA is critical. With the advancement of high-throughput sequencing, exosomal non-coding RNAs (ncRNAs) have emerged as as star molecules, especially with increasing research breakthroughs in CAD diagnosis and treatment. This review primarily discusses the latest advancements in exosomal ncRNAs in CAD diagnosis and treatment.

Overview of exosomal ncRNAs

Production and release of exosomes

Extracellular vesicles (EVs) are membranous particles that are discharged from cells into the external environment. Based on the biogenesis mechanisms of EVs, there are three subtypes: exosomes, microvesicles and apoptotic bodies. Nanovesicles, ranging in size from 30 to 150 nanometers, are known as exosomes, which are derived from the budding of the plasma membrane and the formation of multivesicular endosomes, containing intraluminal vesicles. Exosomes contain various bioactive substances (e.g., nucleic acids, proteins, and lipids), which can be conveyed to recipient cells, facilitating intercellular communication, which bioactive (Lai et al., 2023; Marar, Starich & Wirtz, 2021). Furthermore, exosomes are released by all cell types (e.g., normal cells, tumor cells, fibroblasts, immune cells, adipocytes, T cells, B cells) and found a various body fluids e.g., blood, breast milk, amniotic fluid, and bronchoalveolar lavage fluid (Akers et al., 2013; Kimiz-Gebologlu & Oncel, 2022) (Fig. 2).

Figure 2 Three types of extracellular vesicles.

Apoptotic bodies: 50–5,000 nm, diverse in shape and size, formed during the apoptotic phase of a cell, containing nuclear materials such as various cellular proteins and DNA. Microvesicles: 50–1,000 nm, more uniform in shape, formed by direct fragmentation of the plasma membrane. Exosomes: 30–150 nm, nanosized particles derived from cells, produced by budding of the plasma membrane and formation of multivesicular endosomes containing intraluminal vesicles. Copyright: Figdraw.

Advantages of ncRNAs in exosomes

Compared to ncRNAs present in plasma, exsomal ncRNA offers distinct advantages. The lipid bilayer of exosomes protects ncRNA from enzymatic degradation in bodily fluids, ensuring longer and more stable (Su et al., 2020).

Exosomal ncRNAs have greater sensitivity in predicting CAD. For instance, serum levels of miRNA-208a are less sensitive for acute coronary syndrome (ACS) diagnosis than exosomal miRNA. In heart failure, exosomal miRNA-146a expression is upregulated in patients, while no such association is found with circulating plasma miRNA-146a (Chang et al., 2021; Zhang et al., 2020). However, the quantity of exosomal secretion is influenced by age, showing a negative correlation and acting as an independent factor (Chang et al., 2021). Due to the nature of their spontaneous formation, exosomes are safer than synthetic nanoparticles (Huyan et al., 2020). As such, exosomes have been used as drug carriers, administered via intravenous injection, subcutaneous injection, intraperitoneal injection, and orally. With their stability, biocompatibility, low immunogenicity, and ability to overcome biological barriers, exosomes are also used as potential therapeutics for CAD (Ortega et al., 2020; Wang, Zhao & Xiao, 2019; Zhang, Duan & Bei, 2019). Based on the ability of exosomes to carry nucleic acids (Kimiz-Gebologlu & Oncel, 2022), the ncRNA they carry has been proven to serve as reliable diagnostic markers for CAD, which is significant for the screening of new biomarkers and diagnosis of the disease (Danielson et al., 2018).

Exosomal ncRNAs and CAD

Classification of exosomal ncRNAs

Increasing evidence indicates a close relationship between exosomal ncRNA and CAD. NcRNAs are classified into two categories based on length: long non-coding RNAs (lncRNAs) for sequences longer than 200 nucleotides and small ncRNA for those shorter than 200 nucleotides (Waititu et al., 2020). Exosomes can carry a substantial amount of ncRNAs, such as microRNAs (miRNAs), lncRNAs, and circular RNAs (circRNAs), (De Gonzalo-Calvo & Thum, 2018). ncRNAs carried by cell-derived nanovesicles are called exosomal non-coding RNAs.

Exosomal ncRNAs are involved in a wide range of systemic diseases. For instance, in autoimmune diseases, tumor microvesicles can present tumor antigens to antigen-presenting cells to trigger an immune response (Battisti et al., 2017; Dionisi et al., 2018). In cardiovascular diseases, exosomes from adipose-derived mesenchymal stem cells, by inhibiting endothelial cells miR-342-5p, protect endothelial cells and offer a new strategy for treating ankylosing spondylitis (Wang et al., 2020a). In nephrology, acute kidney injury is a commonly encountered clinical condition (Zarbock et al., 2023). It has been found that exosomes derived from BM-MSCs accelerate renal self-repair post-ischemia-reperfusion in mice, reducing pro-inflammatory cytokines such as IL-6 and TNF-α, and elevating the anti-inflammatory cytokine IL-10 levels (Xie et al., 2022). A growing body of research shows that ncRNA dysregulation is closely linked to the pathophysiology of CAD (Thum & Condorelli, 2015). Specifics will be provided below.

Mechanisms of action of exosomal ncRNAs in CAD

Regulation of inflammatory responses

Exosomal ncRNAs can influence the development of CAD by regulating inflammatory responses. Research has found that exosomal miR-27b-3p from visceral fat can enter vascular endothelial cells, downregulating PPARα and activating the NF-κB pathway, increasing inflammation and atherosclerosis. Conversely, overexpression of PPARα can reduce inflammation and prevent atherosclerosis (Tang et al., 2023).

Promotion of angiogenesis

A specific expression of exosomal circRNAs has been found in the heart during ischemia/reperfusion (I/R) injury, implicating the importance of circRNAs in the pathophysiology of I/R (Ge et al., 2019). Exosomes containing circHIPK3 released from hypoxic cardiomyocytes can be transferred to vascular endothelial cells (ECs). Upon binding to miR-29a, they inhibit the expression of IGF-1, thereby reducing oxidative stress-induced damage and protecting ECs (Wang et al., 2019). Furthermore, exosomal circHIPK3 can inhibit the activity of miR-29a, which in turn promotes the expression of VEGFA. This accelerates the cell cycle and the proliferation of cardiac ECs, promoting angiogenesis and increasing the blood and oxygen supply to the cardiomyocytes. Thus, it ameliorates myocardial ischemia and treats CAD (Wang et al., 2020b).

Regulation of cellular functions

Exosomal ncRNAs can be taken up by target cells, thereby regulating their functions. The pathological characteristic of microscopic polyangiitis (MPA) is vascular inflammation caused by leukocyte infiltration. A study found that exosomes from MPA contribute to cell transfer of miR-1287-5p, promoting the development of acute endothelial injury in MPA and affecting the pathological process of cardiovascular diseases (Zhu et al., 2023).

Mediating intercellular communication

Exosomal ncRNAs are closely associated with CAD. The communication medium of exosomes depends on the lipid bilayer, transferring lipids, proteins, and nucleic acids to adjacent or internal cells (Li et al., 2018). This process mainly occurs in three forms: exosomal surface receptors binding with receptors on the recipient cell membrane, recipient cell internalization of exosomes, and exosomal content release into the cytoplasm following membrane fusion (Liu et al., 2019; Munich et al., 2012; Tkach et al., 2017; Zhang et al., 2023b). Through these processes, exosomes transmit information to other cells, allowing ncRNAs to bind with recipient cells and exert effects by targeting receptors to regulate genes and drive signaling pathways (Ormazabal et al., 2022; Wang et al., 2021a; Zhang et al., 2023a).

Overall, exosomal ncRNAs hold a broad potential in the diagnosis and treatment of CAD. The following sections will detail the role of exosomal ncRNAs in the diagnosis and treatment of CAD.

Exosomal ncRNAs: diagnosis of CAD

Exosomal miRNAs

MiRNAs are a group of small ncRNAs, about 19–25 nucleotides in length, proven to be associated with the pathophysiology of CAD and disease progression. An in vitro experiment in which blood was collected from patients who underwent coronary angiography, serum was separated and exosomes were extracted, analysed by flow cytometry and finally subjected to next-generation sequencing. A clinical study has showed up-regulates circulating exosomal miRNAs including miRNA-382-3p, miRNA-432-5p, miRNA-200a-3p, and miRNA-3613-3p, while down-regulated miRNAs included miRNA-125a-5p, miRNA-185-5p, miRNA-151a-3p, and miRNA -328-3p (Chang et al., 2021). This study found miRNAs associated with CAD, but validating the link between these miRNAs and CAD requires further exploration.

Matrix metalloproteinases (MMPs) can degrade extracellular matrix (ECM) collagen and other structural proteins (Newby, 2008). The ECM plays a crucial role in the pathogenesis of atherosclerosis and cardiovascular diseases. Research found that elevated levels of MMP-9 reflect the rupture of atherosclerotic plaques and myocardial tissue damage (Lahdentausta et al., 2018). Diagnostically, the MMP-9 level and the MMP-9/TIMP-1 molar ratio are associated with ACS (OR 5.81, 95% CI [2.65–12.76], and 4.96, 2.37–10.38), thus serving as early biomarkers. Building on this, Chen et al. (2020) discovered that serum exosomal NEAT1 and MMP expressions in patients with acute ST-segment elevation myocardial infarction are upregulated and positively correlated, with miR-204 expression downregulated, suggesting NEAT1 may affect MMP-9 through miR-204.

Talin-1 (TLN1) gene is one of the major components of the ECM and plays a crucial role not only in the adhesion between integrins and ECM but also in tissue structural remodeling and integrity. Hence, partial rupture of arterial atherosclerotic plaques is associated with downregulation of the TLN1 gene (Davies, 2009; Goult et al., 2010; Goult et al., 2013; Sun et al., 2008; Von Essen et al., 2016). High expression of miR-182-5p and miR-9-5p in CAD has been shown to lead to the downregulation of the TLN1 gene, affecting the interaction between endothelial cells and ECM. This promotes the formation of inflammatory mediators and plaques, thereby compromising vascular integrity (Gholipour et al., 2022; Nieswandt & Watson, 2003; Ruggeri, 2002). Given the significant role of TLN1 in CAD, miR-182-5p and miR-9-5p could be potential biomarkers for CAD.

During the formation of atherosclerosis, the transformation of macrophages into foam cells is related to cholesterol balance regulation. Lectin-like oxidized low-density lipoprotein receptor-1 (LOX-1) is a crucial receptor for binding and internalizing lipoproteins (Kattoor, Goel & Mehta, 2019; Poznyak et al., 2020; Remmerie & Scott, 2018; Tall, Costet & Wang, 2002). An experiment demonstrated that the downstream target of miR-186-5p in serum exosomes from acute myocardial infarction (AMI) patients is LOX-1. Dysregulation of miR-186-5p promotes the development of aortic atherosclerosis, potentially through the upregulation of LOX-1, which affects foam cell formation and enhances lipid absorption in macrophages, exacerbating atherosclerosis (Ding et al., 2022). This study also indicates that miR-186-5p can serve as a biomarker for assessing atherosclerosis.

Exosomal circRNAs

CircRNAs are involved in the formation, proliferation, and differentiation of blood vessels (Newby, 2008). Derived from pre-miRNA (Nieswandt & Watson, 2003), circRNAs are a special type of ncRNA with a covalently closed-loop structure, lacking 5′ to 3′ polarity and a polyadenylated tail. They are formed by back-splicing, creating a closed circular structure (Altesha et al., 2019; Chen & Yang, 2015). Due to their unique circular structure, circRNAs are resistant to exonuclease degradation and are more stable than linear RNAs (Aufiero et al., 2019). CircRNAs were first discovered in the testes of adult mice, yeast mitochondria, and viruses, and they are involved in various diseases, including cardiovascular diseases (Altesha et al., 2019; Arnberg et al., 1980; Capel et al., 1993; Sanger et al., 1976). Currently, about 30,000 different circRNAs have been identified in the human body (Aufiero et al., 2019) and in 1995, it was first discovered that circRNAs could have protein-coding potential (Chen & Sarnow, 1995). Exosomal circRNAs could serve as potential biomarkers for CAD. For example, He et al. (2023) have indicated that has_circRPRD1A and has_circHERPUD2 could act as biomarkers for diagnosing CAD, providing epidemiological support for the interaction between circRNAs and CAD risk factors.

Exosomal ncRNAs: treatment of CAD

Exosomal miRNAs

During the formation of atherosclerosis, activation of vascular smooth muscle cells (VSMCs) and macrophage infiltration promotes plaque formation (Liu et al., 2020). Macrophages have two polarization states: pro-inflammatory (M1) and anti-inflammatory (M2), each playing different roles in various stages of inflammation (Wang, Liang & Zen, 2014). In endothelial cells, high expression of exosomal miRNA-125a-5p inhibits macrophage inflammatory responses by suppressing the NF-κB signaling pathway (Hao et al., 2014; Ormseth et al., 2015). Currently, exosomal miR-21-5-p, miR-126-3p, and miR-100 are known to be associated with atherosclerosis and play crucial roles in function of arterial endothelium (Canfrán-Duque et al., 2017; Gao et al., 2019; Jin et al., 2018; Linna-Kuosmanen et al., 2021; Zhang et al., 2013). MiR-100, for example, inhibits the expression of endothelial cell adhesion molecules and exerts significant anti-inflammatory effects by enhancing autophagy through the MTORC1 signaling pathway(Pankratz et al., 2018). Another study found that exosomal miR-223 from monocytes, stimulated by peonol, decreased the expression of interleukin-1β (IL-1β), interleukin-6 (IL-6), intercellular adhesion molecule-1 (ICAM-1), and vascular cell adhesion moecule-1(VCAM-1) in human umbilical vein endothelial cells (HUVECs), thereby reducing the inflammatory response in coronary artery endothelial cells (Liu et al., 2018).

Circulating exosomes isolated from healthy controls and acute myocardial infarction patients can induce macrophage activation, regulate macrophage polarization, and reduce cardiomyocyte apoptosis, thus protecting the heart from oxidative stress (Zhang et al., 2022). Overexpression of exosomal miR-92a in endothelial cells can inhibit angiogenesis both in vitro and in vivo. In mouse models of coronary ischemia and myocardial infarction, inhibiting exosomal miR-92 has been shown to promote vascular growth and functional recovery in damaged vessels (Bonauer et al., 2009). Shyu et al. (2020) further confirmed this by showing that hyperbaric oxygen induced the expression of lncRNA MALAT1, which suppresses the expression of exosomal miR-92a in a rat model of myocardial infarction, thus promoting angiogenesis and improving the infarcted area. Additionally, Ishii et al. (2006) and others identified a susceptibility locus for myocardial infarction (MI) on chromosome 22q12.1 through a large-scale case-control association study using single nucleotide polymorphisms (SNPs). They discovered a cDNA of new gene within the genome, named myocardial infarction associated transcript (MIAT), and six SNPs of MIAT may confer genetic risk for MI.

In a rat model of coronary heart disease, upregulation of exosomal miR-339 activates the Sirt2/Nrf2/FOXO3 signaling pathway, exacerbating cellular oxidative stress damage. This suggests that the downregulation of exosomal miR-339 has a protective effect against cellular oxidative stress, making it a potential biomarker and therapeutic target for oxidative stress in CAD (Shi et al., 2021).

Since myocardial cells are non-regenerative, myocardial infarction caused by myocardial ischemia results in necrotic heart muscle being replaced by fibrous scar tissue (Frangogiannis, 2019). Fibroblasts differentiate into myofibroblasts, which not only secrete extracellular matrix proteins to maintain cardiac function integrity but also secrete anti-inflammatory factors to reduce inflammation (Ma et al., 2017; Nassiri & Rahbarghazi, 2014). Exosomes secreted from hypoxic bone marrow mesenchymal stem cells (BM-MSCs) can improve myocardial cell apoptosis and promote cardiac repair through exosomal miR-125b in mice with myocardial infarction (Zhu et al., 2018).

Protecting endothelial cells to delay atherosclerosis is important, and it was found that the exosome miR-126-5P is likely to be a novel way to promote endothelial cell recovery and prevent in stenosis after vascular injury, (Mormile, 2020). Additionally, exosomes from adipose-derived mesenchymal stem cells protect endothelial cells and delay the progression of atherosclerosis by inhibiting miR-342-5p (Wang et al., 2020a). These studies indicate that exosomal miRNAs can not only delay the progression of arteriosclerosis through the repair of endothelial cells but also serve as potential biomarkers for the condition.

Exosomal lncRNAs

In a rat model of acute myocardial infarction, exosomes isolated from mesenchymal stem cells (MSC-Exo) compared to those from atorvastatin (ATV) pretreated mesenchymal stem cells (MSCATV-Exo) improved cardiac function recovery, further reduced infarct size, and decreased cardiomyocyte apoptosis. The primary mechanism involved lncRNAH19 in MSCATV-Exo regulating the expression of miR-675 and mediating the activation of vascular endothelial growth factor (VEGF) and intercellular adhesion molecule-1 (Huang et al., 2020). Furthermore, in the plasma exosomes of CAD patients, lnc-MRGPRF-6:1 is highly expressed and positively correlates with the levels of inflammatory cells (tumor necrosis factor-α (TNF-α), tumor necrosis factor-β (TNF-β), and recombinant human C-X-C motif chemokine11 (CXCL11)) in the patient’s plasma. Lnc-MRGPRF-6:1 is upregulated in M1 cells, and upon knockdown of nc-MRGPRF-6:1, the polarization of M1 macrophages decreases. Plasma exosomal lnc-MRGPRF-6:1 promotes macrophage-mediated inflammation by regulating the TLL4-MyD88-MAPK signaling pathway in macrophage M1 polarization (Hu et al., 2022). Bioinformatics analysis indicates that the miR-450a-2-3p/MAPK1 pathway affects cardiac fibrosis. Human pericardial fluid exosomal LINC00636 can counteract cardiac fibrosis and is positively correlated with miR-450a-2-3p. Exosomes containing LINC00636 inhibit MAPK1 by overexpressing miR-450a-2-3p in human pericardial fluid, thereby improving myocardial fibrosis in patients with atrial fibrillation (Liu, Luo & Lei, 2021). This research is crucial for new methods in the prevention and treatment of myocardial fibrosis in atrial fibrillation. These studies demonstrate that exosomal ncRNAs not only can serve as potential biomarkers for CAD but also play a vital role in the treatment of CAD.

Exosomal circRNAs

In CAD, exosomal circ-0001273 derived from umbilical cord mesenchymal stem cells can inhibit cardiomyocyte apoptosis (Li et al., 2020). Exosomal circ-0002113 from mesenchymal stem cells can suppress cell apoptosis after myocardial ischemia-reperfusion (Tian et al., 2021). Exosomal circ-0001747 from adipose-derived stem cells can enhance cell survival and proliferation, as well as inhibit cell inflammation and apoptosis following myocardial ischemia-reperfusion (Duggan et al., 2022). Exosomes containing cPWWP2A and circHIPK3 from umbilical cord mesenchymal stem cells can inhibit the onset of inflammation (Wang et al., 2021b; Yan et al., 2020). Additionally, the circRNA-0006896-miR1264-DNMT1 axis can regulate endothelial cells in atherosclerosis and plays a significant role in atherosclerotic plaques. Culturing HUVECs with exosomes extracted from the serum of unstable plaque atherosclerosis patients showed increased expression of circRNA-0006896, decreased expression of miR-1264, and promoted proliferation and migration of HUVECs (Wen et al., 2021). Furthermore, HUVECs were induced with oxidatively modified oxLDL at various concentrations. Subsequent RT-PCR analysis revealed an upregulation in the expression levels of circ-0003575. However, the silencing of circ-0003575 promoted the proliferation and angiogenesis of oxLDL-induced HUVECs, while concurrently reducing apoptosis in these cells (Li, Ma & Yu, 2017). Liu et al. (2022) found that circ-0026218 attenuates oxLDL-induced inhibitory effects on cell proliferation and apoptosis in HUVECs by regulating the miR-188-3p/TLR4/NF-κB pathway. Xiong et al. (2021) discovered that knocking out circNPHP4 in exosomes derived from monocytes might inhibit heterotypic adhesion of monocytes and coronary artery endothelial cells by reducing miR-1231, potentially through interactions within the circNPHP4/miR1231/EGFR axis. These studies all indicate that reducing inflammatory responses has a protective effect on the development of atherosclerosis (Fig. 3).

Figure 3 Exosomal ncRNAs promote the formation and functional recovery of damaged vessels.

Abbreviations: IL-6, interleukin-6; ICAM-1, intercellular adhesion molecule-1; VCAM-1, vascular cell adhesion moecule-1; LOX-1, low-density lipoprotein receptor-1. Copyright: Figdraw.

Exosomal ncRNAs, as novel biomarkers, have the potential to diagnose CAD. With advancing research, it is believed that exosomal ncRNAs will offer more opportunities for the early diagnosis and treatment of CAD, and provide new insights and clues for related drug targets and therapeutic strategies.

Conclusion

Evaluation of the diagnostic value of exosomal ncRNAs for CAD

For CAD patients, exosomes from coronary blood samples of CAD patients were extracted for next-generation sequencing. The relative expression of exosomal ncRNAs was determined by qRT-PCR. These highly or poorly expressed exosomal non-coding RNAs were further validated and analysed in comparison with normal controls. In the diagnosis of CAD by exosomal miRNAs presented herein, exosomal miRNA-382-3p, miRNA-432-5p, miRNA-200a-3p, and miRNA-3613-3p were highly expressed, miRNA-125a-5p, miRNA-185-5p, miRNA-151a-3p, and miRNA -328-3p were lowly expressed. Although the findings indicated that these exosomal non-coding RNAs are associated with CAD, there are some limitations, and further validation is needed to determine whether there is a chance association. Moreover, these exosomal non-coding RNAs were not analysed by ROC curve analysis to further explore their diagnostic value for CAD, and can only be used as potential biomarkers for CAD.

In STEMI patients, serum exosomal NEAT1 and MMP-9 expression levels were increased, whereas miR-204 expression levels were decreased. All of these non-coding RNAs made predictions of diagnostic value for CAD. However, no comparison was made with the diagnostic value of the traditional myocardial injury markers, high-sensitivity troponin T or CK-MB. However, it is worth affirming that the concentration of MMP-9 is more easily detected than the classical myocardial injury marker high-sensitivity troponin T in patients with CAD, which can respond to the early rupture of plaques and can also predict the occurrence of early acute cardiovascular events.

In patients with CAD, high expression of exosomes miR-182-5p and miR-9-5p can lead to down-regulation of the Talin-1 gene, which affects endothelial cell interactions with the ECM and favours the formation of inflammatory mediators and plaques, thereby undermining vascular integrity. Indirectly reacting to the approximate condition of atherosclerotic plaques by detecting the expression of exosomes miR-182-5p and miR-9-5p in patients with CAD can be used as a potential biomarker for CAD. In AMI patients, exosome miR-186-5p is highly expressed, which not only has high diagnostic value, but also inhibits macrophage atherosclerosis by regulating the downstream molecule LOX-1, which can be used as a biomarker to assess atherosclerosis.

Table 1 Exosomal ncRNAs associated with CAD—May be classified as two types: potential markers and therapeutic targets.

Exosomal ncRNA	Target	Exosome source	Classification of CAD	Significance	Refs.	
miR-92a	Unknown	Animal serum	AMI	Promotes functional recovery of damaged blood vessels and blood vessel growth	Bonauer et al. (2009)	
miRNA-382-3P	Unknown	Coronary artery blood of CAD	Coronary heart disease	Early biomarker	Chang et al. (2021)	
miRNA-432-5P	Unknown	Coronary artery blood of CAD	Coronary heart disease	Early biomarker	Chang et al. (2021)	
miRNA-200a-3P	Unknown	Coronary artery blood of CAD	Coronary heart disease	Early biomarker	Chang et al. (2021)	
miRNA-3613-3P	Unknown	Coronary artery blood of CAD	Coronary heart disease	Early biomarker	Chang et al. (2021)	
miRNA-125a-5p	Unknown	Coronary artery blood of CAD	Coronary heart disease	Early biomarker	Chang et al. (2021)	
miRNA-185-5p	Unknown	Coronary artery blood of CAD	Coronary heart disease	Early biomarker	Chang et al. (2021)	
miRNA-151a-3p	Unknown	Coronary artery blood of CAD	Coronary heart disease	Early biomarker	Chang et al. (2021)	
miRNA328-3p	Unknown	Coronary artery blood of CAD	Coronary heart disease	Early biomarker	Chang et al. (2021)	
NEAT1	miR-204	Patient serum	STEMI	Early biomarker	Chen et al. (2020)	
miR-186-5p	Lox-1	Patient serum	AMI	Affects foam cell formation, promotes macrophage lipid uptake, and exacerbates atherosclerosis	Ding et al. (2022)	
circ-0001747	Unknown	Adipose-derived stem cells	Myocardial ischaemia	Inhibits cellular inflammation and apoptosis	Duggan et al. (2022)	
miR-182-5p	Talin-1 gene	Patient serum	Coronary heart disease	Promotes inflammatory mediators and plaque formation and disrupts
vascular completion	Gholipour et al. (2022)	
miR-9-5p	Talin-1 gene	Patient serum	Coronary heart disease	Promotes inflammatory mediators and plaque formation and disrupts vascular completion	Gholipour et al. (2022)	
miRNA-125a-5p	NF-κB	Patient serum	AMI	Inhibits macrophage inflammatory response	Hao et al. (2014)	
lnc-MRGPRF-6:1	Unknown	Patient plasma	Coronary heart disease	Promotes macrophage- mediated inflammatory responses	Hu et al. (2022)	
circ-0001273	Unknown	Umbilical cord mesenchymal stem cells	Coronary heart disease	Inhibits apoptosis in cardiomyocytes	Li et al. (2020)	
miR-100-5p	mTOR	Pericardial fluid	Myocardial ischaemia	Enhances endothelial cell autophagy and exerts anti-inflammatory effects	Linna-Kuosmanen et al. (2021)	
circ-0026218	miR-188-3p/ TLR4/NF-κB	Patient serum	Atherosclerosis	Attenuates ox-LDL-induced dysfunction in inhibiting cell proliferation and promoting apoptosis	Liu et al. (2022)	
LINC00636	Unknown	pericardial fluid	Atrial fibrillation	Improves myocardial fibrosis in patients with atrial fibrillation	Liu, Luo & Lei (2021)	
miR-182-5p	Talin-1 gene	Patient serum	Coronary heart disease	Promotes inflammatory mediators and plaque formation and disrupts vascular completion	Nieswandt & Watson (2003)	
miR-9-5p	Talin-1 gene	Patient serum	Coronary heart disease	Promotes inflammatory mediators and plaque formation and disrupts vascular completion	Nieswandt & Watson (2003)	
miRNA-125a-5p	NF-κB	Patient serum	AMI	Inhibits macrophage inflammatory response	Ormseth et al. (2015)	
miR-182-5p	Talin-1 gene	Patient serum	Coronary heart disease	Promotes inflammatory mediators and plaque formation and disrupts vascular completion	Ruggeri (2002)	
miR-9-5p	Talin-1 gene	Patient serum	Coronary heart disease	Promotes inflammatory mediators and plaque formation and disrupts vascular completion	Ruggeri (2002)	
miR-339	Sirt2/Nrf2/ FOXO3	Patient serum	Coronary heart disease	Exacerbates oxidative stress damage to cells	Shi et al. (2021)	
lncRNA MALAT1	Unknown	Cardiomyocyte	AMI	Promotes vessel growth and improves infarct size	Shyu et al. (2020)	
circ-0002113	Unknown	Mesenchymal stem cell MSC	Myocardial ischaemia	Inhibits apoptosis	Tian et al. (2021)	
circHIPK3	Unknown	Umbilical cord mesenchymal stem cells	Myocardial ischaemia	Inhibits the inflammatory response	Wang et al. (2021b)	
circ-0006896	Unknown	Patient serum	Atherosclerosis	Promotes the proliferation and migration of human umbilical vein endothelial cells	Wen et al. (2021)	
circNPHP4	circNPHP4/ miR1231/ EGFR axis	Monocyte	Atherosclerosis	Inhibits heterogeneous adhesion of monocytes and coronary endothelial cells	Xiong et al. (2021)	
circHIPK3	Unknown	Umbilical cord mesenchymal stem cells	Myocardial ischaemia	Inhibits the inflammatory response	Yan et al. (2020)	

Exosomes has_circRPRD1A and has_circHERPUD2 were down-regulated in CAD expression and correlated with established risk factors for CAD (age, gender, hypertension, diabetes), among others. Exosomal has_circRPRD1A (AUC = 0.689) and has_circHERPUD2 (AUC = 0.662), although diagnostic value for CAD, were not compared with the diagnostic value of classical high-sensitivity troponin T. This result seems that the exosomal has_circRPRD1A and has_circHERPUD2 diagnostic value is not satisfactory (Table 1).

Evaluation of the therapeutic value of exosomal non-coding RNA in CAD

For the therapeutic efficacy of exosomal non-coding RNA in CAD, exosomal non-coding delays the onset of atherosclerosis by inhibiting the inflammatory response of arterial endothelial cells. It protects cardiomyocytes by reducing cardiomyocyte apoptosis, resisting myocardial fibrosis, promoting cardiomyocyte survival and proliferation, and promoting the growth and functional recovery of damaged blood vessels in infarcted mice. However, their therapeutic potential for CAD, which is currently not applied in the clinic, is illustrated by cellular and animal studies. It is believed that we will see the application and therapeutic effects of exosomes in specific diseases in the future.

With increasing research into exosomes, the value of exosomal ncRNAs in the early diagnosis of CAD is evident. However, the pathway and protective mechanisms of exosomal ncRNAs in diseases, supported by solid experimental validation are poorly defined. We also face huge technical challenges on how exosomal ncRNAs can be specifically applied to clinical diagnosis and treatment of CAD, including exosome production, isolation, loading efficiency, biological distribution, and absorption. These challenges are further complexed by additional confounding diseases, such as lung cancer, chronic renal failure, and endocrine diseases since these confounding diseases may affect the predictive value of potential markers for CAD. Much more research is needed to develop the clinical application of ncRNAs for the diagnosis and treatment of CAD.

Additional Information and Declarations

Competing Interests

Author Contributions

Data Availability

The authors declare there are no competing interests.

Cuixue Ma conceived and designed the experiments, performed the experiments, analyzed the data, prepared figures and/or tables, authored or reviewed drafts of the article, and approved the final draft.

Xinyu Gao performed the experiments, analyzed the data, prepared figures and/or tables, authored or reviewed drafts of the article, and approved the final draft.

Kongwei Li performed the experiments, analyzed the data, prepared figures and/or tables, and approved the final draft.

Yin Liu performed the experiments, analyzed the data, prepared figures and/or tables, and approved the final draft.

Yuan He conceived and designed the experiments, authored or reviewed drafts of the article, and approved the final draft.

Yang Zhang analyzed the data, authored or reviewed drafts of the article, and approved the final draft.

Wei Lei conceived and designed the experiments, analyzed the data, authored or reviewed drafts of the article, and approved the final draft.

The following information was supplied regarding data availability:

This is a literature review.

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
