# Peer review of "Exosomal non-coding RNAs: key molecules in the diagnosis and treatment of coronary artery disease"

_PeerJ, doi:10.7717/peerj.19352_

## Round 0.1 · original submission · Minor Revisions

The authors are requested to carefully revise the manuscript and answer the questions raised by the reviewers.

Reviewer 1 ·

Basic reporting

1.Keywords: Non-coding RNA actually includes the MicroRNA, LncRNA and CircRNA. I suggest deleting the MicroRNA, LncRNA and CircRNA in the Keywords.
2.Introduction: I suggest adding the description of what exosomal non-coding RNA is in lines 99-102 to provide the reader with a better understanding. Additionally, the role of exosomal non-coding RNA in other disease also should be described briefly
3.In the section of Production and release of exosomes, in addition to body fluid exosomes, I suggest adding the introduction of tissue exosomes
4.In Table 1, I suggest adding the source of exosomes and specific classification of CAD. Also, please check the correspondence between the references in Table 1 and the contents of the table.
5.Need to improve language in this review, e.g. lines 157-158 use “studies are proving…” for a study published in 2015.
6.It is not clear what the cellular components in Figure 2 represent, and I suggest adding the appropriate textual annotations.
7.Try to use original research rather than reviews in references
8.The conclusion is too long, it is recommended to shorten the text and try to avoid citing references.

Experimental design

no comment

Validity of the findings

no comment

Additional comments

no comment

Reviewer 2 ·

Basic reporting

The English language used in this article is smooth and unambiguous.

Experimental design

Check the detail comments at the Section 4.

Validity of the findings

Check the detail comments at the Section 4.

Additional comments

This article aims to discuss and summarize the role of non-coding RNA in coronary heart disease and its contributions to diagnosis and treatment. While this topic is interesting, enthusiasm is lessened as there was not a critical review of the literature and in many cases, conclusions were not drawn. Outstanding issues are noted below:
There is a lack of critical review of the literature presented in literature. Additionally, the cited studies do not seem to be comprehensive. For instance, Chandrasekera D, (Cardiovasc Diabetol, 2022 Jul 1) have summarized the role of exosomal miRNA in cardiovascular disease, Li J. et al (Epigenomics. 2022 Nov) have reported the function of both non-exosomal and exosomal ncRNAs in the regulation of CAD, etc. However, these parts/data are not mentioned in the current manuscript.
Some references in the manuscript simply list data without the necessary analysis and summary, which are the most important part of a review. For example, line 200-205, the authors only cited the sentence from another review without providing any more summary or conclusions. What type of these studies (in vivo, in vitro, or clinical data)? What are the strengths and weaknesses of the data obtained from these experiments? What is the principle of the experiment? …etc.

Other minor issues such as:
The reference 28 in the Table 1 is incorrect.
Section 2 “Search strategy” is not necessary.
The Table is too simplistic.

·

Basic reporting

It´s a very good job of researching with all the types of ncRNA in exosomes in coronary artery disease. If I have to say something, you have to specify all the acronyms with the meaning, as with NEAT1 or AMI, that for a researcher in the field of cardiology it may be obvious but for a researcher in another field it is not.

Experimental design

It's a well structured job.

Validity of the findings

Since the work is based on a review of previously published works, I have nothing to comment.

---

## Round 0.2 · accepted · Accept

After revisions, one reviewer agreed to publish the manuscript. There are two reviewers left with major revisions, and I think the author has responded adequately. I also reviewed the manuscript and found no obvious risks to publication. Therefore, I also approved the publication of this manuscript.